# Nicotine treatment regulates PD-L1 and PD-L2 expression via inhibition of Akt pathway in HER2-type breast cancer cells

**Masanori A. Murayama**[1,2], **Erika Takada**[1], **Kenji Takai**[1], **Nagisa Arimitsu**[1], **Jun Shimizu**[1], **Tomoko Suzuki**[1], **Noboru Suzuki**[1] *

**1** Department of Immunology and Medicine, St. Marianna University of School of Medicine, Miyamae-ku, Kawasaki, Kanagawa, Japan, **2** Department of Animal Models for Human Diseases, Institute of Biomedical Science, Kansai Medical University, Hirakata, Osaka, Japan

* n3suzuki@marianna-u.ac.jp

**Data Availability Statement:** All relevant data are within the manuscript.

## Abstract

The immune checkpoint molecules such as PD-L1 and PD-L2 have a substantial contribution to cancer immunotherapy including breast cancer. Microarray expression profiling identified several molecular subtypes, namely luminal-type (with a good-prognosis), HER2-type (with an intermediate-prognosis), and triple-negative breast cancer (TNBC)-type (with a poor-prognosis). We found that PD-L1 and PD-L2 mRNA expressions were highly expressed in TNBC-type cell lines (HCC1937, MDA-MB-231), moderately expressed in HER2-type cell line (SK-BR-3), and poorly expressed in luminal-type cell lines (MDA-MB-361, MCF7). The PD-L1 and PD-L2 expression in SK-BR-3 cells, but not those in HCC1937 and MDA-MB-231 cells, decreased by nicotine stimulation in a dose-dependent manner. In addition, nicotine treatment decreased the phosphorylation of Akt in SK-BR-3 cells, but not in other cell lines. These results show that nicotine regulates the expression of immune checkpoint molecules, PD-L1 and PD-L2, via inhibition of Akt phosphorylation. This findings may provide the new therapeutic strategies for the treatment of breast cancer.

## Introduction

Immune-checkpoint molecules, such as programmed cell death protein 1 (PD-1), programmed death ligand 1 (PD-L1) and PD-L2 are outstanding targets for cancer immunotherapy [1]. PD-1 is particularly expressed on cytotoxic T cells. PD-L1 is ubiquitously expressed in many tissues and cells including dendritic cells and PD-L2 expression is restricted to macrophages and dendritic cells [2]. The binding of PD-1 on T cells to PD-L1 and PD-L2 on antigen-presenting cells negatively regulates T cell effector function [3]. The PD-L1 and PD-L2 expressed on tumor cells cause tumor immune escape [4].

Recently, breast cancer was categorized into several subtypes such as luminal-type (estrogen receptor (ER) and progesterone receptor (PR) positive, good-prognosis), HER2-type (human epidermal growth factor receptor 2 positive, intermediate prognosis), and triple-negative breast cancer-type (TNBC, ER, PR and HER2 negative, poor-prognosis) [5]. The levels of PD-L1 expression were different in each subtype of breast cancer cell lines [6]. The PD-L1 expression was associated with histological grade, pathological stage, tumor infiltrating

**Funding:** This work was supported in part by SRF (Tokyo, Japan, https://www.srf.or.jp/) to NS. The funders had no role in study design, data collection and analysis, decision to publish, or preparation of the manuscript.

**Competing interests:** The authors have declared that no competing interests exist.

lymphocytes (TILs), better disease-free survival (DFS), and overall survival (OS) in breast cancer patients. Based on the intrinsic subtype, PD-L1 expression associates with the levels of TILs in HER2- and TNBC-type patients [7]. Another report also showed that the PD-L1 expression associates with histological grade, TILs, and DFS in HER2-type breast cancer [8]. As with PD-L1 expression in breast cancer patients, PD-L2 expression is positive in half of the breast cancer patients. However, the PD-L2 expression is not associated with better OS [9].

Smoking is one of risk factors in breast cancer [10]. Nicotine is a natural compound in tobacco plants, which is a highly addictive. Nicotine exerts biological effects on excitable and non-excitable cells via nicotinic acetylcholine receptors (nAChRs) [11]. The expression levels of nAChR subunits are different on breast cancer cell lines [12]. It is reported that nicotine acts on breast cancer cells such as luminal- and TNBC-type via some nAChRs [13–21]. However, the nicotine-mediated biological effects including the regulation of immune-check point molecules on HER2-type breast cancer cells are largely unknown. In this study, we focused on the nicotine-mediated expression of immune-check point molecules, PD-L1 and PD-L2, on breast cancer cells using luminal-, HER2-, and TNBC-type breast cancer cell lines.

## Materials and methods

### Cell lines

Human breast cancer cell lines; luminal-type (MDA-MB-361 and MCF7), HER2-type (SK-BR-3) and TNBC-type (HCC1937, MDA-MB-231) were purchased from American Type Culture Collection (ATCC, Rockville, MD, USA). According to manufacturer's instructions, MDA-MB-361 cells were cultured in in Leibovitz's L-15 medium (ATCC) containing 20% FBS (without heat inactivation) without $CO_2$ aeration. MCF7 cells were cultured in Eagle's Minimum Essential Medium (ATCC) containing 10% FBS (without heat inactivation) and 0.01 mg/ml of human recombinant insulin under the 5% $CO_2$ aeration. SK-BR-3 cells were cultured in McCoy's 5a Medium (ATCC) containing 10% FBS (without heat inactivation) under the 5% $CO_2$ aeration. HCC1937 cells were cultured in RPMI-1640 medium (ATCC) containing 10% FBS (without heat inactivation) under the 5% $CO_2$ aeration. MDA-MB-231 cells were cultured in Leibovitz's L-15 medium (ATCC) containing 10% FBS (without heat inactivation) without $CO_2$ aeration.

### Nicotine treatment

All cell lines ($6 \times 10^4$ cells/well) were seeded in 24-well plate (Iwaki, Shizuoka, Japan) and cultured with or without nicotine (SIGMA, St. Louis, MO, USA) for 24 h.

### Real-time PCR

Total RNA was extracted using RNeasy Plus Mini Kit (Qiagen, Valencia, CA, USA) from $1 \times 10^5$ cells of breast cancer cell lines in 24-well plate and reverse transcribed using High-Capacity cDNA Reverse Transcription Kit (Applied Biosystems, Foster City, CA, USA). We performed real-time PCR using TaqMan Gene Expression Master Mix (Applied Biosystems) or THUNDERBIRD SYBR qPCR Mix (TOYOBO) and 7300 Real-Time PCR System (Applied Biosystems) with a set of primers which were purchased from Thermo Fisher Scientific (Waltham, MA, USA, *ACTB*: Hs01060665_g1, *PDL1*: Hs00204257-m1, and *PDL2*: Hs00228839-m1) and were described in Table 1 [22–24].

### Immunohistochemistry

Cytospin slides ($2 \times 10^4$ cells/slide) were prepared using NewSilane II Micro Slides (Muto Pure Chemicals, Tokyo, Japan), fixed with 4% PFA at room temperature for 10 min, blocked with

**Table 1. The real-time PCR primers sets.**

| Gene name | Forward primer | Reverse primer |
|---|---|---|
| CHRNA1 | GCTCTGTCGTGGCCATCAA | CCGGAAAGCGACCAGCCAGA |
| CHRNA2 | GTGGAGGAGGAGGACAGA | CTTCTGCATGTGGGGTGATA |
| CHRNA3 | CAGAGTCCAAAGGCTGCAAG | AGAGAGGGACAGCACAGCAT |
| CHRNA4 | CTCACCGTCCTTCTGTGTC | CTGGCTTTCTCAGCTTCCAG |
| CHRNA5 | CTTCACACGCTTCCCAAACT | CTTCAACAACCTCACGGACA |
| CHRNA6 | TCCATCGTGGTGACTGTGT | AGGCCACCTCATCAGCAG |
| CHRNA7 | GTACGCTGGTTTCCCTTTGA | CCACTAGGTCCCATTCTC |
| CHRNA9 | GAAAGCAGCCAGGAACAAAG | GCACTTGGCGATGTACTCAA |
| CHRNA10 | ACACAAGTGCCCTGAGACCT | TCCCATCGTAGGTAGGCATC |
| CHRNB1 | CTACGACAGCTCGGAGGTCA | GCAGGTTGAGAACCACGACA |
| CHRNB2 | GGCATGTACGAGGTGTCCTT | CACCTCACTCTTCAGCACCA |
| CHRNB3 | AACAGTTCCGTTTGATTTCACGAT | CCCTGATGACCAAGGTCATC |
| CHRNB4 | TCCCTGGTCCTTTTCTTCCT | TGCAGCTTGATGGAGATGAG |
| CHRNG | CGCCTGCTCTATCTCAGTCA | GGAGACATTGAGCACAACCA |
| CHRND | CAGATCTCCTACTCCTGCAA | CCACTGATGTCTTCTCACCA |
| CHRNE | TCAAGGTCACCCTGACGAAT | GTCGATGTCGATCTTGTTGA |
| KLF4 | GAAATTCGCCCGCTCCGATGA | CTGTGTGTTTGCGGTAGTGCC |
| WNT5A | CTTCGCCCAGGTTGTAATTGAAGC | CTGCCAAAAACAGAGGTGTTATCC |

Blocking One Histo (Nacalai Tesque, Kyoto, Japan) at room temperature for 2 h, and then stained with primary antibodies to human PD-L1 (1:200 dilution, #ab205921, Abcam, Cambridge, MA, USA), PD-L2 (1:500, #MAB1224, R&D Systems, Minneapolis, MN, USA), Akt (1:200, #9272, Cell Signaling, Danvers, MA, USA) and phospho-Akt (1:200, #ab105731, Abcam) at 4˚C for overnight, coupled with secondary antibodies to Alexa Fluor® 594 goat anti-rabbit IgG (1:300, #A11072, Thermo Fisher Scientific) and anti-mouse IgG (1:300, #A11005, Thermo Fisher Scientific) at room temperature for 3h. For nucleus staining, we used DAPI (0.4 μg/ml, #D9564, SIGMA). The immunofluorescence was examined with a confocal microscope (LSM-800, Zeiss, Oberkochen, Germany) and staining intensity was quantitated using ImageJ software (National Institutes of Health, Bethesda, MD, USA).

### Cell viability assay

SK-BR-3 cells were seeded at a density of $1 \times 10^4$ cells/well on 96-well plate. After 24 h culture, the cells were treated with nicotine (0, 1, 10, and 100 nM) for 24 h. Cell viability was measured by MTT (3-[4, 5-dimethylthiazol-2-yl]-2, 5 diphenyl tetrazolium bromide) assay reagent (Nacalai, Japan) according to manufacturer's protocol.

### Statistical analysis

Two-sided Student's $t$-test was performed for all statistical evaluation. $P<0.05$ was considered as statistically significant. Data are expressed as mean ± standard error of the mean (SEM).

## Results

### PD-L1 and PD-L2 mRNA expressed in HER2-type and TNBC-type cell lines

PD-L1 and PD-L2 expressions were found in about half of breast cancer cases [9]. Consistent with previous report [6], we found that PD-L1 mRNA highly expressed in HER2-type SK-BR-3 cells, TNBC-type HCC1937 cells, and MDA-MB-231 cells (Fig 1A).

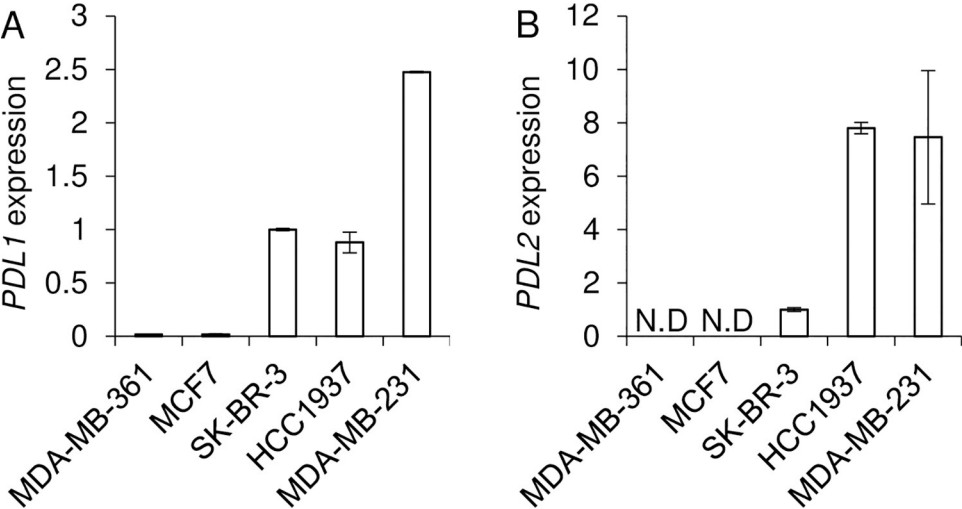

**Fig 1. The mRNA expression of PD-L1 and PD-L2 in breast cancer cell lines.** The relative mRNA expressions of
PD-L1 (A) and PD-L2 (B) on breast cancer cell lines were measured by qPCR (n = 4 each). The values were relative to
the mRNA expressions of SK-BR-3. Each mean ± SEM is shown. N.D: Not detected.

It is reported that TNBC-type cell lines expressed PD-L1 and PD-L2 [25]. In addition to
TNBC-type cells (HCC1937 and MDA-MB-231), we found that PD-L2 mRNA was expressed
in SK-BR-3 cells (Fig 1B).

## Nicotine treatment decreased PD-L1 and PD-L2 expressions via inhibition of Akt pathway in a HER2-type cell line

To elucidate the role of nicotine in the expressions of PD-L1 and PD-L2 in breast cancer cells,
the mRNA expressions were evaluated after nicotine treatment in the molecule positive cell
lines. The PD-L1 mRNA expressions in HCC1937 and MDA-MB-231 cells were not affected
by nicotine treatment (Fig 2A and 2B). We found that nicotine treatment decreased PD-L1
mRNA (Fig 2C) and protein (Fig 2D and 2E) expressions in SK-BR3 cells in a dose dependent
manner.

PD-L2 mRNA expressions in HCC1937 and MDA-MB-231 cells were not affected by nico-
tine treatment (Fig 3A and 3B). We found that nicotine treatment decreased PD-L2 mRNA
(Fig 3C) and protein (Fig 3D and 3E) expressions in SK-BR3 cells in a dose dependent
manner.

Cell proliferation, cell migration, and maintenance of cancer stem cell features are impor-
tant for tumorigenesis. We found that nicotine treatment did not affect the cell viability of
SK-BR-3 cells (Fig 4A). Next, we investigated the expression of KLF4, which is important for
the maintenance of breast cancer stem cell features and the promotion of the cell migration
and invasion [22]. Nicotine stimulation did not influence KLF4 mRNA expression in SK-BR-3
cells (Fig 4B). Wnt5a is important for cell migration [26]. Nicotine treatment decreased Wnt5a
mRNA expression in SK-BR-3 cells in a dose-dependent manner (Fig 4C). It suggests that nic-
otine treatment inhibits breast cancer cell migration.

Wnt5a signaling induces cancer cell migration via Akt phosphorylation [27,28]. PD-L1 and
PD-L2 was expressed via Akt pathway in cancer including breast cancer [29–31]. We exam-
ined the role of nicotine treatment for Akt phosphorylation in SK-BR-3 cells. The phosphory-
lation of Akt decreased at 30 min after nicotine treatment in the cell line (Fig 5A). On the
other hand, we found that MCF7, HCC1937 and MDA-MB-231 cells did not express Akt

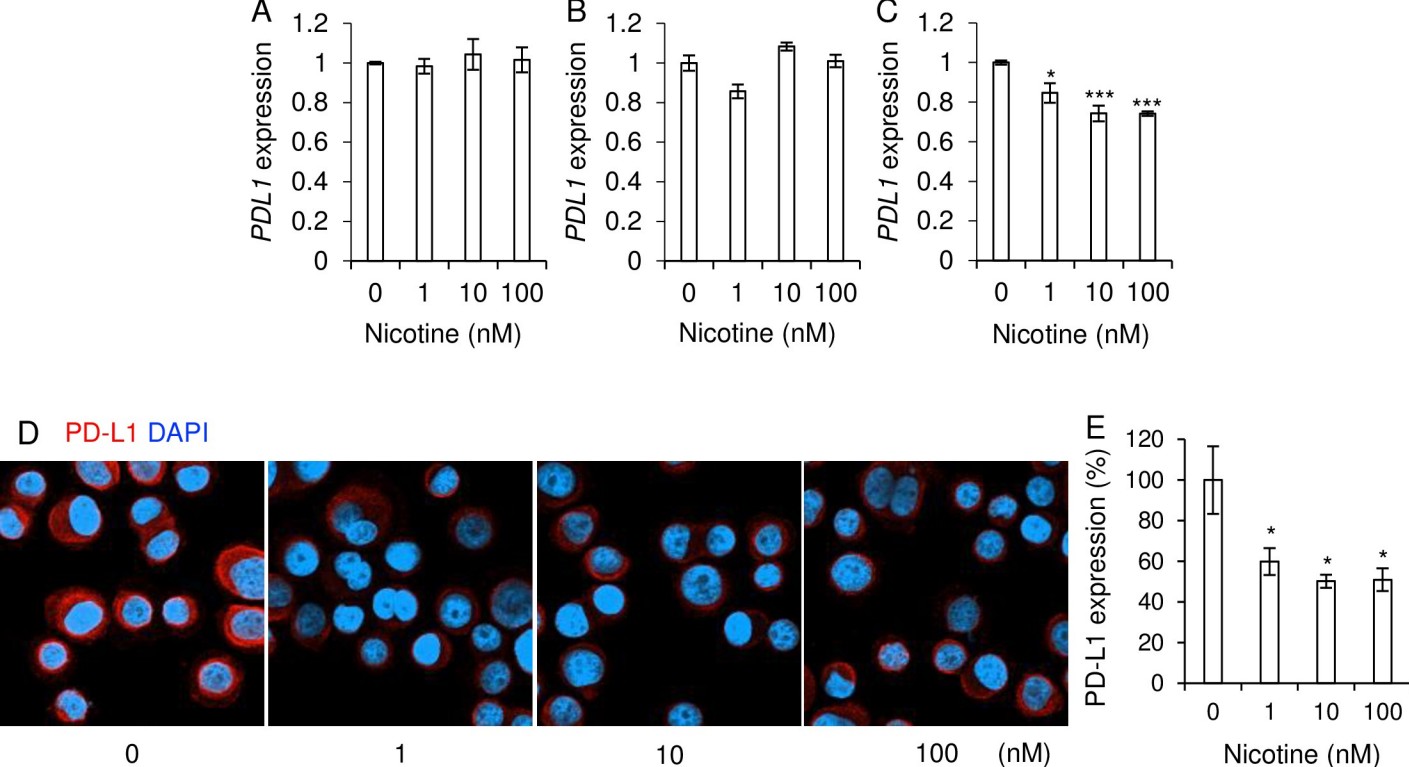

**Fig 2. The effect of nicotine treatment for PD-L1 expressions on breast cancer cell lines.** (A-C) Breast cancer cell lines, HCC1937 cells (A), MDA-MB-231 cells (B), and SK-BR-3 cells (C) were stimulated with nicotine (0–100 nM) and the relative mRNA expressions of PD-L1 to those without the stimulations (nicotine 0 nM: Relative expression = 1) were measured by qPCR (n = 4 each). (D, E) SK-BR-3 cells were cultured in the absence and presence of nicotine (1, 10 and 100 mM) and were stained with anti-PD-L1 antibody. DAPI was used for detecting nuclei. (D) One of the representative figures is shown. (E) The relative expression of PD-L2 was determined by using ImageJ (n = 5 each). Each mean ± SEM is shown. $^*p < 0.05$ and $^{***}p < 0.001$.

protein and nicotine treatment did not affect Akt expression and the phosphorylation in these cells (Fig 5B–5D). As expected, an Akt inhibitor, MK-2206, suppressed PD-L1 and PD-L2 expressions in a dose-dependent manner (Fig 5E and 5F). Thus, these results show that nicotine treatment affects PD-L1 and PD-L2 expressions via inhibition of Akt pathway in a HER2-type cell line.

Nicotine exerts biological effects via various combinations of nAChR subunits [11]. We examined mRNA expressions of several nAChR subunits in breast cancer cell lines by real-time PCR (Fig 6). We did not detect nAChRβ3, γ, δ and ε mRNA expressions in breast cancer cell lines. SK-BR-3 cells expressed substantially several nAChR subunits mRNA compared with other subtype breast cancer cells except for nAChRα1 and α4 subunits.

## Discussion

In this study, we found that immune-check point molecules, PD-L1 and PD-L2, were highly expressed in TNBC-type HCC1937 and MDA-MB-231 cells and were moderately expressed in HER2-type SK-BR-3 cells. The PD-L1 and PD-L2 expressions were decreased by nicotine treatment via inhibition of Akt phosphorylation in SK-BR-3 cells, but not in HCC1937 and MDA-MB-231 cells. Thus, nicotine and related molecules can be useful therapeutic targets in HER2-type breast cancer for the cancer immunotherapy.

Nicotine exerts biological effects on excitable and non-excitable cells via various combinations of nAChR subunits [11]. 16 homologous genes encode the subunits of nAChR. Muscle-

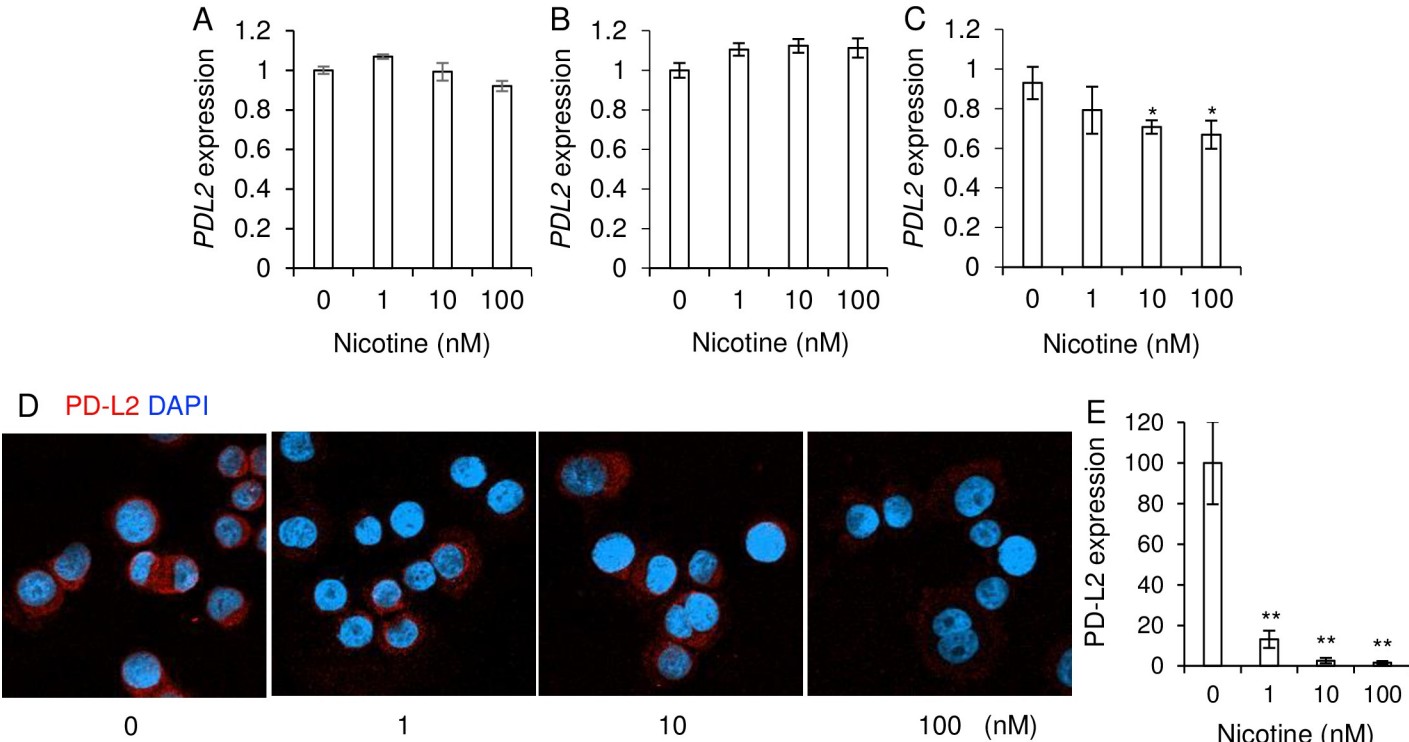

**Fig 3. The effect of nicotine treatment for PD-L2 expressions on breast cancer cell lines.** (A-C) Breast cancer cell lines, HCC1937 cells (A), MDA-MB-231 cells (B), and SK-BR-3 cells (C) were stimulated with nicotine (0–100 nM) and the relative mRNA expressions of PD-L2 to those without the stimulations (nicotine 0 nM: Relative expression = 1) were measured by qPCR (n = 4 each). (D, E) SK-BR-3 cells were cultured in the absence and presence of nicotine (1, 10 and 100 mM) and were stained with anti-PD-L2 antibody. DAPI was used for detecting nuclei. (D) One of the representative figures is shown (n = 5 each). (E) The relative expression of PD-L2 was determined by using ImageJ. Each mean ± SEM is shown. *p < 0.05 and **p < 0.01.

type nAChRs include nAChR(α1)₂β1δε (adult receptor) or nAChR(α1)₂β1δγ (fetal receptor). Neuronal-type nAChRs are homo- or hetero-pentamers composed of some of nine α subunits (α2-α10) and some of three β subunits (β2–4) [11]. Each combination of nAChR subunits has distinct and specific roles in the biological processes of muscles and neurons. The expression profile of nAChR subunits is different in subtypes of breast cancer [12]. The nAChRα5

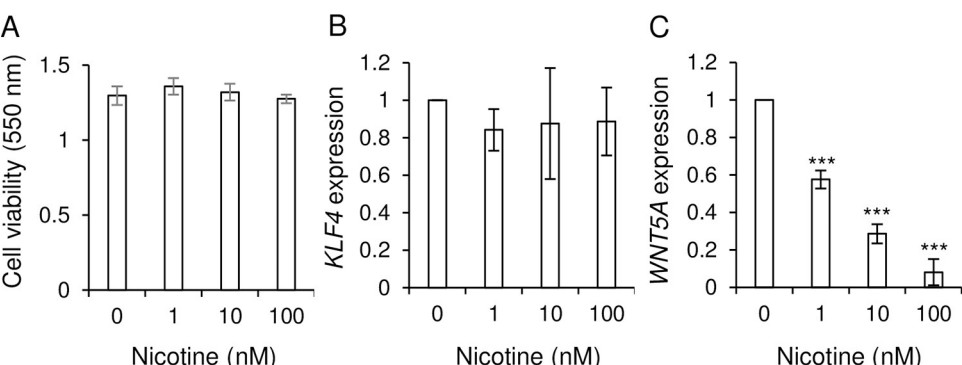

**Fig 4. Nicotine decreased Wnt5a expressions in SK-BR-3 cells.** (A) Cell viability of SK-BR-3 cells with nicotine treatment (0–100 nM) was measured by MTT assay (n = 4 each). (B, C) KLF4 (B) and Wnt5a (C) mRNA expressions in SK-BR-3 cells at 24 h after nicotine treatment were measured by qPCR (n = 4 each). Each mean ± SEM is shown. ***p < 0.001.

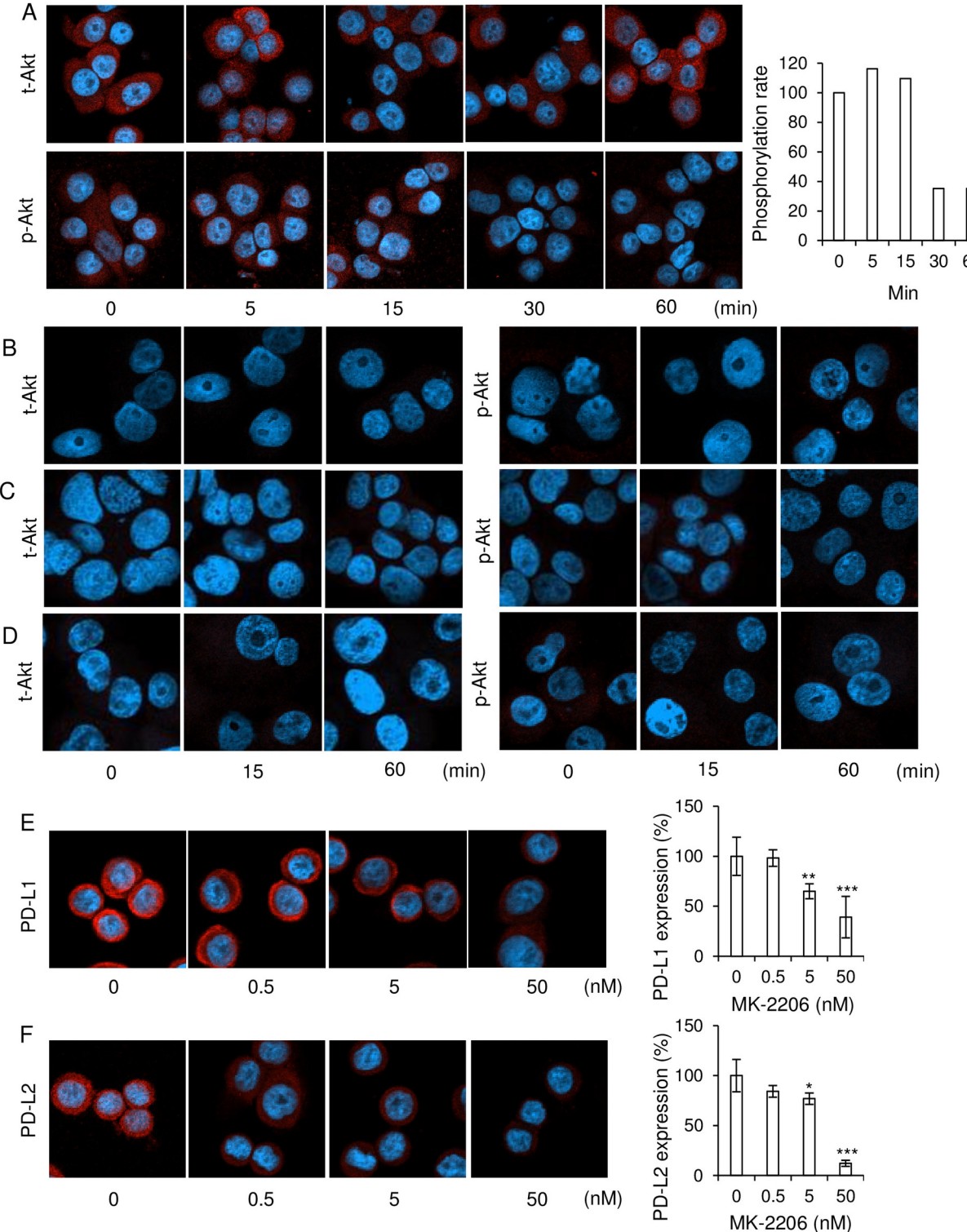

**Fig 5. Nicotine decreased Akt phosphorylation in SK-BR-3 cells.** (A) Akt expressions and the phosphorylation in nicotine-treated SK-BR-3 cells were assayed by immunostaining. SK-BR-3 cells were stimulated with 100 nM nicotine for indicated time and the cells were stained with indicated antibodies (red). DAPI (blue) was used for detecting nuclei. Some of representative figures of total-Akt (upper), phosphorylated Akt (lower) are shown. The relative phosphorylation intensity of Akt was calculated using ImageJ (n = 5–7 each). (B-D) Akt expressions and the phosphorylation in nicotine-treated MCF8 (B), HCC1937 (C) and MDA-MB-231 (D) cells were assayed by immunostaining. SK-BR-3 cells were stimulated with 100 nM nicotine for indicated time and the cells were stained with indicated antibodies (red). DAPI (blue) was used for

detecting nuclei. Some of representative figures of total-Akt (left), phosphorylated Akt (right) are shown (n = 3–4 each). (E, F) The PD-L1 (E) and PD-L2 (F) expressions in MK-2206, an Akt inhibitor, -treated SK-BR-3 cells are shown. SK-BR-3 cells were stimulated with MK-2206 (0–50 nM) for 24 h and the cells were stained with indicated antibodies (red). DAPI (blue) was used for detecting nuclei. Some of representative figures of PD-L1 or PD-L2 (left) and the relative expression levels (right) are shown. The relative expression levels of PD-L1 or PD-L2 are determined by using ImageJ (n = 5–7 each).

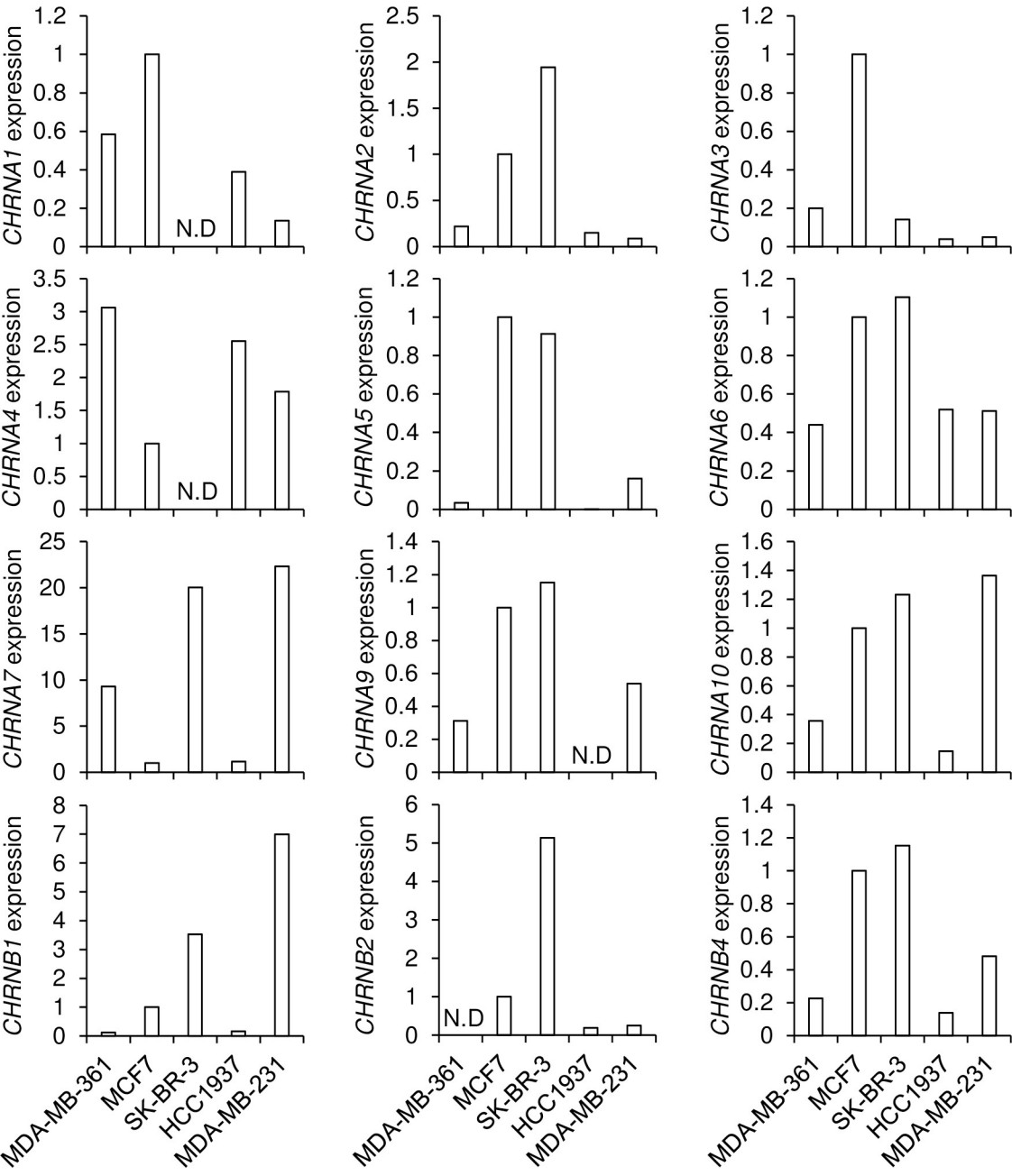

**Fig 6. The expression of nAChR subunits in breast cancer cells.** The relative mRNA expressions of nAChR subunits in breast cancer cell lines was measured by qPCR. The values were relative to the mRNA expressions of MCF7 cells. N.D.: Not detected.

associates with cell cycle, apoptosis, and DNA damage response of luminal-type breast cancer cells [13]. The nAChRα7 promotes cell proliferation and apoptosis in a panel of breast cancer cell lines [14]. The nAChRα9 promotes invasion ability, apoptosis resistance, and growth of luminal-type breast cancer cells [15–18]. The nAChRα9 promotes growth and transformation of TNBC-type breast cancer cells [19,20]. The nonselective nAChR antagonist inhibits nicotine-induced TNBC-type breast cancer cell growth [21]. Thus, nicotine causes negative effects for cancer immunotherapy on luminal- and TNBC-type breast cancers via nAChRα5, α7 and α9 subunits. Furthermore, nAChRα9 mediates nicotine-induced PD-L1 expression in melanoma cells [32]. However, nicotine treatment decreased PD-L1 and PD-L2 expression in HER2-type breast cancer cells in this study. We found that SK-BR-3 cells expressed substantially several nAChR subunits mRNA compared with other subtype breast cancer cells except for nAChRα1 and α4 subunits. It suggested that these nAChR subunits mediated nicotine-induced cancer immunotherapy on HER2-type breast cancer cells.

The expression of PD-L1 and PD-L2 is regulated by Akt pathway on cancer cells including breast cancer [29–31]. Furthermore, interaction of PD-1 and PD-L1 causes resistance to chemotherapy via activation of Akt pathway [33]. Thus, Akt-targeted therapy is important to overcome multi-drug resistance in breast cancer [34]. We found that nicotine decreased Akt phosphorylation on HER2-type cells. In contrast, luminal- and TNBC cells did not express Akt and nicotine treatment did not affect Akt expressions in these cells. These results suggest that nicotine may exert positive effects for the cancer immunotherapy of HER2-type cancers through different mechanisms from the negative effects on the luminal- and TNBC-type breast cancers. This finding suggests that nicotine treatment could develop useful therapeutic methods for HER2-type breast cancer.

## Author Contributions

**Conceptualization:** Masanori A. Murayama, Jun Shimizu, Tomoko Suzuki, Noboru Suzuki.

**Data curation:** Masanori A. Murayama, Tomoko Suzuki.

**Formal analysis:** Masanori A. Murayama, Erika Takada.

**Funding acquisition:** Tomoko Suzuki, Noboru Suzuki.

**Investigation:** Masanori A. Murayama, Erika Takada, Kenji Takai, Nagisa Arimitsu.

**Project administration:** Tomoko Suzuki, Noboru Suzuki.

**Resources:** Kenji Takai, Nagisa Arimitsu.

**Supervision:** Tomoko Suzuki, Noboru Suzuki.

**Validation:** Noboru Suzuki.

**Visualization:** Masanori A. Murayama, Erika Takada.

**Writing – original draft:** Masanori A. Murayama, Jun Shimizu, Noboru Suzuki.

**Writing – review & editing:** Masanori A. Murayama, Jun Shimizu, Noboru Suzuki.

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
