## [Decision Letter · Decision Letter 0]

17 Jun 2021

PONE-D-21-09905

Nicotine treatment regulates PD-L1 and PD-L2 expression via inhibition of Akt pathway in HER2-type breast cancer cells

PLOS ONE

Dear Dr. Suzuki,

Thank you for submitting your manuscript to PLOS ONE. After careful consideration, we feel that it has merit but does not fully meet PLOS ONE’s publication criteria as it currently stands. Therefore, we invite you to submit a revised version of the manuscript that addresses the points raised during the review process.

From the reviews provided, it can be seen that a significant amount of additional molecular/mechanistic data is needed for this to be of interest to PLOS One.  As indicated by both the reviewers, the study is premature and does not fully support the conclusions or provide a cohesive, complete story.  A significant amount of additional molecular data would be necessary to make this a convincing story and this would be the expectation, if you were to submit a revised version.

We look forward to receiving your revised manuscript.

Kind regards,

Srikumar Chellappan

Academic Editor

PLOS ONE

Journal Requirements:

"This work was supported in part by Smoke Research Foundation (SRF, Tokyo,

Japan, https://www.srf.or.jp/) to NS . The funders had no role in study design,

data collection and analysis, decision to publish, or preparation of the

manuscript."

"NO"

Reviewers' comments:

Reviewer's Responses to Questions

**Comments to the Author**

1. Is the manuscript technically sound, and do the data support the conclusions?

Reviewer #1: No

Reviewer #2: Partly

2. Has the statistical analysis been performed appropriately and rigorously? 

Reviewer #1: No

Reviewer #2: No

3. Have the authors made all data underlying the findings in their manuscript fully available?

Reviewer #1: Yes

Reviewer #2: Yes

4. Is the manuscript presented in an intelligible fashion and written in standard English?

Reviewer #1: Yes

Reviewer #2: Yes

5. Review Comments to the Author

Reviewer #1: In this manuscript, authors investigated the expression patterns of immune checkpoint molecules PDL1 and PDL2 in different subtypes of breast cancer cell lines. Authors show that nicotine treatment was able to reduce the expression of PDL1 and PDL2 in HER2 positive breast cancer cell line SK-BR-3 and showed no inhibitory effect in TNBC cell line HCC1937 and MDA-MB-231. Further, authors have observed that nicotine mediated inhibitory effect on PDL1 and PDL2 in HER2 positive breast cancer cell line SK-BR-3 was dependent on the regulation of Akt pathway and may provide the new therapeutic strategies for the treatment of breast cancer. The data reported here appears to be very preliminary and more in-depth work is required to improve the scope of this manuscript.

1. It is important to investigate the effect of nicotine treatment mediated changes in the PD-L1/L2 expression and the impact on cancer stem cell like properties, proliferation, and migration.

2. nAChR subunits expression levels in these cell lines should be included and its correlation with PD-L1 and PD-L2 needs to be investigated.

3. In addition to qRT-PCR data of mRNA expression, authors should validate the PD-L1 surface expression of PD-L1 and PD-L2 by flow cytometry in all cell lines before and after nicotine treatment.

4. Figure 2D, please provide better quality of immunofluorescence images for PDL1 and also provide immunofluorescence images for PDL2 in figure 2E.

5. Overall, the quality of immunofluorescence images is not convincing

6. AKT phosphorylation with one cell line is not enough. Authors need to perform in multiple cell lines. It also lacks proper positive control with AKT inhibitor to look at PDL1 and PDL2 expression. Or consider knocking down of Akt expression by siRNA experiments.

Reviewer #2: Reviewer’s Comment:

The present study by Murayama MA et al titled “Nicotine treatment regulates ………. Breast cancer cells” ineptly demonstrated a very preliminary rather a qualitative observation where an active natural tobacco ingredient, nicotine suppresses immune checkpoint molecules, PD-L1 and PD-L2 expression in HER2+ type breast cancer cells which in turn might correlate directly or indirectly with the dephosphorylation status of AKT molecules in these types of cells. Given that the basic concept of AKT phosphorylation depends on the status of the immune checkpoint molecule, PD-Ls expression that majorly studied in the field of gastric cancers, nevertheless, in-depth molecular analysis in the field of breast cancer, especially on this particular subtype (HER2+ in this case) might be of tremendous interest in the field and warrants more comprehensive and mechanistic studies. In a nutshell, the current study is a premature and trivial piece of work without much in-depth molecular analysis and hence does not attract this reviewer in favor of publication. The areas to improve in the study are listed below,

1. How PD-L expressions directly regulate AKT phosphorylation that could be investigated in the breast cancer cells, especially in the context of HER2+ subtypes followed by the modulation upon nicotine administration. Authors should investigate the molecular insight of it. The presented data is just the tip of the iceberg!

2. A single cell line data can always be deceptive and therefore, needs supplementation and/or recapitulation in few more cell lines under a particular subtype before drawing any major conclusion.

3. What is the nicotinic acetylcholine receptor(s) status under the different subtypes of breast cancer cells utilized in the study? A comprehensive RT-PCR panel would have been interesting to correlate the effect of nicotine in these subtypes under the study.

4. To establish the real correlation between PD-L1 and PD-L2 expression with phospho-AKT and nAChRs expression, authors could have done comprehensive ICC/IHC studies on commercially available BC tissue microarrays (TMAs) and could have expanded their analysis on the different subtypes in BC!

5. How nicotine differentially modulates the AKT phosphorylation via modulation of PD-L expression between HER2+ versus TNBC cells though they both express these same set of immune checkpoint molecules? This fundamental question also remains unanswered in the manuscript.

6. PLOS authors have the option to publish the peer review history of their article (what does this mean?). If published, this will include your full peer review and any attached files.

Reviewer #1: **Yes: **Krithika Kodumudi

Reviewer #2: **Yes: **SahaB

---

## [Author Response · Author response to Decision Letter 0]

28 Aug 2021

Correspondence to Editor

 Thank you for reviewing our manuscript entitled "Nicotine treatment regulates PD-L1 and PD-L2 expression via inhibition of Akt pathway in HER2-type breast cancer cells". Now we are submitting the revised manuscript. We really appreciate your kind comments and, according to your suggestion, we have revised the manuscript as follows.

[Journal Requirements]

Thank you for your comments. We revised manuscript at your style requirements.

2. Thank you for stating the following in the Acknowledgments Section of your manuscript: "This work was supported in part by Smoke Research Foundation (SRF, Tokyo, Japan, https://www.srf.or.jp/) to NS . The funders had no role in study design,

data collection and analysis, decision to publish, or preparation of the manuscript." We note that you have provided funding information that is not currently declared in your Funding Statement. However, funding information should not appear in the Acknowledgments section or other areas of your manuscript. We will only publish funding information present in the Funding Statement section of the online submission form. Please remove any funding-related text from the manuscript and let us know how you would like to update your Funding Statement. Currently, your Funding Statement reads as follows: "NO". Please include your amended statements within your cover letter; we will change the online submission form on your behalf.

Thank you for your comments. We removed the funding-related text from the manuscript and we described the statement in the cover letter according to your comment. 

Correspondence to Reviewer #1

In this manuscript, authors investigated the expression patterns of immune checkpoint molecules PDL1 and PDL2 in different subtypes of breast cancer cell lines. Authors show that nicotine treatment was able to reduce the expression of PDL1 and PDL2 in HER2 positive breast cancer cell line SK-BR-3 and showed no inhibitory effect in TNBC cell line HCC1937 and MDA-MB-231. Further, authors have observed that nicotine mediated inhibitory effect on PDL1 and PDL2 in HER2 positive breast cancer cell line SK-BR-3 was dependent on the regulation of Akt pathway and may provide the new therapeutic strategies for the treatment of breast cancer. The data reported here appears to be very preliminary and more in-depth work is required to improve the scope of this manuscript. 

Thank you for reviewing our manuscript. We revised our manuscript according to your comments and suggestions as follows.

1. It is important to investigate the effect of nicotine treatment mediated changes in the PD-L1/L2 expression and the impact on cancer stem cell like properties, proliferation, and migration. 

We evaluated PD-L1 (Fig. 2)/PD-L2 (Fig. 3) expressions with/without nicotine treatment by qPCR and IHC. And according to your suggestions, we investigated the impact of nicotine treatment on the cell proliferation using MTT assay. We found that nicotine treatment did not alter the proliferation rates of SK-BR-3 cells (Fig. 4A).

We investigated the expression of KLF4, which is important for the maintenance of breast cancer stem cell features and the promotion of the cell migration and invasion (Ref. 22:Oncogene. 2011 May 5;30(18):2161-72.). The mRNA expression of KLF4 did not change by nicotine stimulation (Fig. 4B). Wnt5a is important for cell migration (Ref 26: Oncotarget. 2018 Apr 20;9(30):20979-20992.). Nicotine stimulation decreased the mRNA expression of Wnt5a in SK-BR-3 cells (Fig. 4C). These results suggested that nicotine treatment had impacts on the cell migration, but not on the cell proliferation and stem cell-like property in SK-BR-3 cells.

2. nAChR subunits expression levels in these cell lines should be included and its correlation with PD-L1 and PD-L2 needs to be investigated. 

According to your suggestion, we investigated several nAChR subunit expressions by qPCR. We observed substantial expressions of several nAChR subunits in SK-BR-3 cells compared with other subtypes of cancer cells. We suggest that nicotine treatment may reduce the expressions of PD-L1 and PD-L2 through these nAChR subunits.

3. In addition to qRT-PCR data of mRNA expression, authors should validate the PD-L1 surface expression of PD-L1 and PD-L2 by flow cytometry in all cell lines before and after nicotine treatment. 

We performed flow cytometry analyses with several antibodies mainly in SK-BR-3 cells and found that the frequencies of PD-L1 expressing cells were 2.0 - 20%. However, the reproducibility of the cell frequencies was limited. Alternatively, we assessed mRNA expressions by qPCR and protein expressions by IHC and demonstrated the results in Fig2 and 3.

4. Figure 2D, please provide better quality of immunofluorescence images for PDL1 and also provide immunofluorescence images for PDL2 in figure 2E.

5. Overall, the quality of immunofluorescence images is not convincing. 

According to your suggestion, we provided better quality images in the revised Figures (Fig. 2D, Fig. 3D, and Fig. 5).

6. AKT phosphorylation with one cell line is not enough. Authors need to perform in multiple cell lines. It also lacks proper positive control with AKT inhibitor to look at PDL1 and PDL2 expression. Or consider knocking down of Akt expression by siRNA experiments. 

We assessed Akt phosphorylation in SK-BR-3, MCF8, HCC1937, and MDA-MB-231 cells (Figs. 5A-D). We observed the phosphorylation only in SK-BR-3 cells. We used an Akt inhibitor, MK-2206, to reduce the expression of PD-L1 and PD-L2 in SK-BR-3 cells (Figs. 5E, F). As expected, the inhibitor suppressed PD-L1 and PD-L2 expression of the cells. Wnt5a signaling was reported to enhance Akt phosphorylation. We found that nicotine treatment decreased Wnt5a mRNA expression (Fig. 4C). Wnt5a may associate with Akt phosphorylation in SK-BR-3 cells. 

Reviewer #2: Reviewer’s Comment:

The present study by Murayama MA et al titled “Nicotine treatment regulates ………. Breast cancer cells” ineptly demonstrated a very preliminary rather a qualitative observation where an active natural tobacco ingredient, nicotine suppresses immune checkpoint molecules, PD-L1 and PD-L2 expression in HER2+ type breast cancer cells which in turn might correlate directly or indirectly with the dephosphorylation status of AKT molecules in these types of cells. Given that the basic concept of AKT phosphorylation depends on the status of the immune checkpoint molecule, PD-Ls expression that majorly studied in the field of gastric cancers, nevertheless, in-depth molecular analysis in the field of breast cancer, especially on this particular subtype (HER2+ in this case) might be of tremendous interest in the field and warrants more comprehensive and mechanistic studies. In a nutshell, the current study is a premature and trivial piece of work without much in-depth molecular analysis and hence does not attract this reviewer in favor of publication. The areas to improve in the study are listed below,

Thank you for reviewing our manuscript. We revised our manuscript according to your comments and suggestion as follows.

1. How PD-L expressions directly regulate AKT phosphorylation that could be investigated in the breast cancer cells, especially in the context of HER2+ subtypes followed by the modulation upon nicotine administration. Authors should investigate the molecular insight of it. The presented data is just the tip of the iceberg! 

Wnt5a was reported to induce breast cancer cell migration via Akt phosphorylation (Ref. 26: Oncotarget. 2018; 9: 20979-92.). And the Akt pathway is suggested to be a therapeutic target for tumor immunity. We found that nicotine treatment decreased Wnt5a mRNA expression and Akt phosphorylation in SK-BR-3 cells. However, MCF7, HCC1937 and MDA-MB-231 cells did not express Akt regardless of the presence and absence of nicotine treatment. These results suggested that nicotine treatment inhibited PD-L1 and PD-L2 expression via inhibition of Akt phosphorylation in SK-BR-3 cells. 

2. A single cell line data can always be deceptive and therefore, needs supplementation and/or recapitulation in few more cell lines under a particular subtype before drawing any major conclusion. 

According to your suggestion, we investigated Akt phosphorylation using other cell lines in Fig.5. We did not find Akt expressions or the phosphorylation regardless of the presence and absence of nicotine treatment in MCF7, HCC1937 and MDA-MB-231 cells in this study. We found clear differences in terms of the Akt pathway activation between SK-BR-3 cells and other types of cancer cells. 

3. What is the nicotinic acetylcholine receptor(s) status under the different subtypes of breast cancer cells utilized in the study? A comprehensive RT-PCR panel would have been interesting to correlate the effect of nicotine in these subtypes under the study. 

In this review process, we investigated several nAChR subunit mRNA levels in the breast cancer cell lines (Fig. 6). We observed substantial expressions of several nAChR subunits in SK-BR-3 cells compared with other subtypes of cancer cells. We suggest that nicotine treatment may reduce the expressions of PD-L1 and PD-L2 through these nAChR subunits.

4. To establish the real correlation between PD-L1 and PD-L2 expression with phospho-AKT and nAChRs expression, authors could have done comprehensive ICC/IHC studies on commercially available BC tissue microarrays (TMAs) and could have expanded their analysis on the different subtypes in BC! 

We performed IHC of Akt and phopho-Akt and demonstrated the results in Fig. 5. We did not find Akt expressions or the phosphorylation regardless of the presence and absence of nicotine treatment in MCF7, HCC1937 and MDA-MB-231 cells. In this study, we would like to discuss the distinct features of the SK-BR-3 cell line in the immune check-point regulation, especially through the Akt pathway, compared with other types of breast cancer cell lines.

5. How nicotine differentially modulates the AKT phosphorylation via modulation of PD-L expression between HER2+ versus TNBC cells though they both express these same set of immune checkpoint molecules? This fundamental question also remains unanswered in the manuscript. 

We found that TNBC cells expressed strongly mRNA of PD-L1 and PD-L2. Nicotine treatment did not affect the expressions. The TNBC cells did not express Akt and phopho-Akt regardless of the presence and absence of nicotine treatment in this study. We found that SK-BR-3 cells expressed moderately mRNA of PD-L1 and PD-L2. Nicotine treatment reduced the expressions. The HER2 type cancer cells expressed phospho-Akt and nicotine treatment decreased the expression. An Akt inhibitor reduced PD-L1 and PD-L2 expressions in SK-BR-3 cells. Nicotine treatment decreased Wnt5a mRNA expression in SK-BR-3 cells. Reduced Wnt5a signaling may regulate Akt phosphorylation in the cells. These results suggested that nicotine treatment suppressed PD-L1 and PD-L2 expression via inhibition of Akt phosphorylation only in SK-BR-3 cells.

---

## [Decision Letter · Decision Letter 1]

18 Nov 2021

Nicotine treatment regulates PD-L1 and PD-L2 expression via inhibition of Akt pathway in HER2-type breast cancer cells

PONE-D-21-09905R1

Dear Dr. Suzuki,

We’re pleased to inform you that your manuscript has been judged scientifically suitable for publication and will be formally accepted for publication once it meets all outstanding technical requirements.

Kind regards,

Srikumar Chellappan

Academic Editor

PLOS ONE

Additional Editor Comments (optional):

Reviewers' comments:

Reviewer's Responses to Questions

**Comments to the Author**

1. If the authors have adequately addressed your comments raised in a previous round of review and you feel that this manuscript is now acceptable for publication, you may indicate that here to bypass the “Comments to the Author” section, enter your conflict of interest statement in the “Confidential to Editor” section, and submit your "Accept" recommendation.

Reviewer #1: All comments have been addressed

Reviewer #2: (No Response)

2. Is the manuscript technically sound, and do the data support the conclusions?

Reviewer #1: Yes

Reviewer #2: Partly

3. Has the statistical analysis been performed appropriately and rigorously? 

Reviewer #1: Yes

Reviewer #2: No

4. Have the authors made all data underlying the findings in their manuscript fully available?

Reviewer #1: Yes

Reviewer #2: Yes

5. Is the manuscript presented in an intelligible fashion and written in standard English?

Reviewer #1: No

Reviewer #2: Yes

6. Review Comments to the Author

Reviewer #1: All comments were addressed by authors satisfactorily. Revised manuscript has included all the key experiments suggested.

Reviewer #2: The revised manuscript did not address all the questions raised by this reviewer, satisfactorily!

1. A single cell line data under a specific subtype of BC which also determines the mechanistic insights, can always be deceptive. This reviewer is skeptical about it.

2. Further, this reviewer is not convinced with the answers made by the authors against question numbers 2, 4, and 5.

7. PLOS authors have the option to publish the peer review history of their article (what does this mean?). If published, this will include your full peer review and any attached files.

Reviewer #1: **Yes: **Krithika Kodumudi

Reviewer #2: No

---

## [Editor Report · Acceptance letter]

18 Jan 2022

PONE-D-21-09905R1 

Nicotine treatment regulates PD-L1 and PD-L2 expression via inhibition of Akt pathway in HER2-type breast cancer cells 

Dear Dr. Suzuki:

I'm pleased to inform you that your manuscript has been deemed suitable for publication in PLOS ONE. Congratulations! Your manuscript is now with our production department. 

Kind regards, 

on behalf of

Dr. Srikumar Chellappan 

Academic Editor

PLOS ONE